| Open Peer Review | *Clinical Microbiology* | *Commentary*

# What's old is new: leveraging existing antimicrobial susceptibility test methods for rapid results in patients with bloodstream infections

Andrea M. Prinzi[1]

**ABSTRACT**  The use of rapid disk diffusion or modified automated antimicrobial susceptibility testing (AST) system approaches demonstrates excellent performance for gram-negative organisms directly from blood cultures. In a recent study, S. Khan, A. Das, A. Mishra, A. Vidyarthi, et al. (Microbiol Spectr 12:e03081-23, 2024, https://doi.org/10.1128/spectrum.03081-23) compared the performance of three direct-from-blood AST methods against standard of care disk diffusion and automated AST. The results demonstrated high categorical agreements and low error rates across three protocols. The study suggests that locally validated direct-from-blood AST protocols offer reliable and fast results, particularly for resource-limited settings. However, local context and workflows should be considered prior to implementing rapid AST protocols, and more research is needed on the performance of rapid AST protocols for gram-positive organisms.

**KEYWORDS**  rapid tests, susceptibility testing, blood culture, gram-negative bacteria, bloodstream infections

The bloodstream infection diagnosis and treatment process has changed significantly over the years, with one key tenet remaining the same: healthy blood is vital to human survival. From blood-letting to leeches and eventually modern methods, one thing is clear—drastic methods have always been taken to treat patients with bloodstream infections as quickly as possible. Sepsis, which is often caused by bloodstream infections, is a life-threatening infection that is responsible for millions of deaths globally each year (1). Studies have demonstrated that the timing of early and appropriate antibiotics is imperative for saving lives in sepsis cases, with the odds of mortality increasing significantly with each hourly delay in therapy (2). The diagnostic landscape for performing antimicrobial susceptibility testing (AST) from blood cultures has morphed significantly over the last several decades, with special attention paid to decreasing the time to AST results. While many test options and methods are currently available for AST from blood cultures (3), the decision of what to use relies heavily on local context, availability of resources and cost, laboratory infrastructure, and clinical need. Given this, understanding the performance and applicability of various methodologies is important so that different settings can choose the most realistic and feasible option for them that will positively impact patient care.

In their recent study, Khan et al. (4) assessed the performance of three direct AST methods compared to reference disk diffusion and standard automated AST. The study was performed in the clinical microbiology laboratory of the National Cancer Institute at the Jhajjar campus of All India Institute of Medical Sciences in New Delhi, India. Positive blood cultures from hematological and solid organ malignancy patients were subjected to various AST protocols, and the performance of each was assessed. The

Address correspondence to Andrea M. Prinzi, Andrea.prinzi@biomerieux.com.

Andrea Prinzi is a paid employee of bioMerieux (Field Medical Director, US Medical Affairs).

*The views expressed in this article do not necessarily reflect the views of the journal or of ASM.*

See the original article at https://doi.org/10.1128/spectrum.03081-23.

first method, protocol A, utilized the Clinical and Laboratory Standards Institute (CLSI) direct-from-blood-culture disk diffusion method (5). Protocol B involved harvesting a bacterial pellet from a centrifuged positive blood specimen and using that pellet to inoculate automated AST panels, which were incubated in and read by the Phoenix AST system (Becton Dickinson, Franklin Lakes, USA). Protocol C leveraged the same centrifuged specimen to perform disk diffusion testing.

Overall, all three methods performed excellently compared to the respective standard of care tests. Among the Enterobacterales, protocols A, B, and C demonstrated categorical agreements of 97.2%, 96.1%, and 96.3%, respectively. Error rates were low and within acceptable limits for all test methods. A comparison of protocol B to standard automated AST yielded 96.1% categorical agreement and 9.5% essential agreement for Enterobacterales organisms. For *Pseudomonas aeruginosa*, protocol A demonstrated 100% categorical agreement.

In comparison, protocols B and C had categorical agreements of 93.5% and 94.7%, respectively, for *P. aeruginosa* and other non-fermenting gram-negative bacilli using the standard of care disk diffusion at the comparator. When standard automated AST was the reference method, non-fermenting gram-negative bacilli tested with protocol B had an overall categorical agreement and essential agreement of 98.2% and 97.6%, respectively. There were no statistical differences among all three protocols and the overall performance when testing Enterobacterales and non-fermenters.

The CLSI direct-from-blood disk diffusion method was developed as a standardized approach to rapid AST in response to more stringent laboratory regulations that limited how labs in the United States could perform off-label testing with their Food and Drug Administration-cleared tests (e.g., setting up positive blood samples on automated AST devices directly, rather than using isolates from solid media) (3). While this method has demonstrated good performance in the original validation study and this present study, the authors note that the number of antibiotics that can be tested using this method is limited. The excellent performance of protocols B and C, combined with performance data from existing literature, suggests that locally validated direct-from-blood AST protocols may be suitable for labs that need to test more antibiotics but have cost or resource limitations. This is especially true for centers that do not have access to automated AST platforms. Notably, the ability of labs to perform off-label testing for rapid AST from blood cultures in the United States may be limited by recent regulatory requirements that place greater restrictions on laboratory-developed testing (6).

Innovation in the field of rapid AST for blood cultures is exploding. Many novel and sophisticated tests are in development or have come to market in recent years, combining various technology types to identify susceptibility patterns both phenotypically and genotypically. In most cases, these new tests involve automated platforms and promise to deliver AST results within 3–7 hours with good performance (3). While these assays may significantly improve clinical management for patients with bloodstream infections, more data on implementation, impact on clinical outcomes, and cost-effectiveness are needed. In some areas of the world, a lack of infrastructure or test costs may be significant barriers to adopting new rapid AST technologies. However, new technologies are not the only ones with limitations that need consideration. In their study, Khan et al. note that the primary limitation to rapid disk diffusion methods is that an organism identification must still be obtained before interpretation, and this can add hours or days to the turnaround time, depending on the identification method used. Additionally, the authors note that more research is needed on the performance of these direct methods for gram-positive organisms. The variability in test options and local opportunity for test implementation remind us of the importance of local context when selecting and assessing tests for use in clinical microbiology labs (7).

Given the severity of bloodstream infections and the high risk of mortality in patients who get them, particularly those with immunocompromising conditions, a reliable and rapid AST method should be accessible across all setting types. As Dr. Paul Farmer, renowned medical anthropologist and humanitarian, once said, "no one should have to

die of a disease that is treatable." Performance data for all test types must be available to those hoping to use them to save lives. The findings from Khan et al., paired with existing evidence, demonstrate the excellent performance of three direct AST options that offer results a day sooner and may support the early administration of optimal therapy in patients with bloodstream infections.

## AUTHOR AFFILIATION

[1]bioMérieux, US Medical Affairs, Salt Lake City, Utah, USA

## AUTHOR ORCIDs

Andrea M. Prinzi  http://orcid.org/0000-0003-2949-5484

## AUTHOR CONTRIBUTIONS

Andrea M. Prinzi, Conceptualization, Methodology, Writing – original draft, Writing – review and editing

## ADDITIONAL FILES

The following material is available online.

### Open Peer Review

**PEER REVIEW HISTORY (review-history.pdf).** An accounting of the reviewer comments and feedback.

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
