## [Reviewer comments · Microbiology Spectrum]

Microbiology Spectrum

What's Old is New: Leveraging Existing Antimicrobial Susceptibility Test Methods for Rapid Results in Patients with Bloodstream Infections

Andrea Prinzi

Corresponding Author(s): Andrea Prinzi, bioMerieux Inc

Review Timeline:

Submission Date:	May 8, 2024
Editorial Decision:	May 28, 2024
Revision Received:	May 28, 2024
Accepted:	June 6, 2024

Editor: Kessendri Reddy

Reviewer(s): The reviewers have opted to remain anonymous.

Transaction Report:

DOI: <https://doi.org/10.1128/spectrum.00995-24>

Re: Spectrum00995-24 (What's Old is New: Leveraging Existing Antimicrobial Susceptibility Test Methods for Rapid Results in Patients with Bloodstream Infections)

Dear Dr. Andrea M Prinzi:

Thank you for the privilege of reviewing your work. Below you will find my comments, instructions from the Spectrum editorial office, and the reviewer comments.

Minor comments:

- Line 62: Change "AST plates" to "AST panels"
- Line 69: The original manuscript text states that the categorical agreement of Protocol B compares with standard automated AST was 96.1% for the Enterobacterales, not 99.1%.

Revision Guidelines

Sincerely,
Kessendri Reddy
Editor
Microbiology Spectrum

28 May 2024

Dear Kessendri Reddy,

I hereby submit the revised manuscript Spectrum00995-24 entitled: "What's Old is New: Leveraging Existing Antimicrobial Susceptibility Test Methods for Rapid Results in Patients with Bloodstream Infections" for publication in Spectrum. I sincerely appreciate the opportunity to improve the quality of this manuscript and appreciate your review. Below, I have listed each comment and corresponding response.

Review comments and responses

Minor comments:

- **Line 62: Change "AST plates" to "AST panels"**

This change has been made at line 62

- **Line 69: The original manuscript text states that the categorical agreement of Protocol B compares with standard automated AST was 96.1% for the Enterobacterales, not 99.1%.**

This change has been made at line 69

Re: Spectrum00995-24R1 (What's Old is New: Leveraging Existing Antimicrobial Susceptibility Test Methods for Rapid Results in Patients with Bloodstream Infections)

Dear Dr. Andrea M Prinzi:

Your manuscript has been accepted, and I am forwarding it to the ASM production staff for publication. Your paper will first be checked to make sure all elements meet the technical requirements. ASM staff will contact you if anything needs to be revised before copyediting and production can begin. Otherwise, you will be notified when your proofs are ready to be viewed.

Sincerely,
Kessendri Reddy
Editor
Microbiology Spectrum